# Interleukin-3-Receptor-α in Triple-Negative Breast Cancer (TNBC): An Additional Novel Biomarker of TNBC Aggressiveness and a Therapeutic Target

**DOI:** 10.3390/cancers14163918

**Published:** 2022-08-13

**Authors:** Malvina Koni, Isabella Castellano, Emilio Venturelli, Alessandro Sarcinella, Tatiana Lopatina, Cristina Grange, Massimo Cedrino, Saveria Femminò, Paolo Cossu-Rocca, Sandra Orrù, Fabrizio D’Ascenzo, Ilaria Cotellessa, Cristian Tampieri, Carla Debernardi, Giovanni Cugliari, Giuseppe Matullo, Giovanni Camussi, Maria Rosaria De Miglio, Maria Felice Brizzi

**Affiliations:** 1Department of Medical Sciences, University of Turin, 10126 Turin, Italy; 2Anatomic Pathology Unit, Department of Diagnostic Services, “Giovanni Paolo II” Hospital, ASL Gallura, 07026 Olbia, Italy; 3Department of Pathology, “A. Businco” Oncologic Hospital, ARNAS Brotzu, 09121 Cagliari, Italy; 4Department of Medical, Surgical and Experimental Sciences, University of Sassari, 07100 Sassari, Italy

**Keywords:** interleukin-3/interleukin-3 receptor α, triple-negative breast cancer, vascular mimicry, programmed cell death-ligand 1

## Abstract

**Simple Summary:**

Molecular and histological profiling is crucial for biomarker and therapeutic target discovery, for example, in TNBC. We demonstrated that IL-3Rα expression led to the identification of a subgroup of TNBC patients displaying a poor overall survival. Moreover, we refined TNBC molecular annotation and drew a model including IL-3Rα, PD-L1, and genes related to EMT, which finely discriminates cancer aggressiveness. Finally, we first demonstrated that IL-3Rα is instrumental in granting tumour adaptation and progression by reprogramming TNBC cells to form large dysfunctional vessels and reshaping PD-L1 expression in primary tumours and metastases. Therefore, the IL-3/IL-3Rα axis may be proposed as a marker of TNBC aggressiveness, as a novel TNBC therapeutic challenge.

**Abstract:**

Tumour molecular annotation is mandatory for biomarker discovery and personalised approaches, particularly in triple-negative breast cancer (TNBC) lacking effective treatment options. In this study, the interleukin-3 receptor α (IL-3Rα) was investigated as a prognostic biomarker and therapeutic target in TNBC. IL-3Rα expression and patients’ clinical and pathological features were retrospectively analysed in 421 TNBC patients. IL-3Rα was expressed in 69% human TNBC samples, and its expression was associated with nodal metastases (*p* = 0.026) and poor overall survival (hazard ratio = 1.50; 95% CI = 1.01–2.2; *p* = 0.04). The bioinformatics analysis on the Breast Invasive Carcinoma dataset of The Cancer Genome Atlas (TCGA) proved that IL-3Rα was highly expressed in TNBC compared with luminal breast cancers (*p* = 0.017, *p*adj = 0.026). Functional studies demonstrated that IL-3Rα activation induced epithelial-to-endothelial and epithelial-to-mesenchymal transition, promoted large blood lacunae and lung metastasis formation, and increased programmed-cell death ligand-1 (PD-L1) in primary tumours and metastases. Based on the TCGA data, IL-3Rα, PD-L1, and EMT coding genes were proposed to discriminate against TNBC aggressiveness (AUC = 0.86 95% CI = 0.82–0.89). Overall, this study identified IL-3Rα as an additional novel biomarker of TNBC aggressiveness and provided the rationale to further investigate its relevance as a therapeutic target.

## 1. Introduction

Triple-negative breast cancer (TNBC) is a subtype of breast cancer characterised by the lack of oestrogen, progesterone, and human epidermal growth factor receptor-2 (HER2) [1]. TNBC mostly arises in young women and accounts for approximately 10–15% of all breast cancers [2]. TNBC is generally a highly aggressive metastatic carcinoma, displaying a high mortality rate and recurrence [3]. Moreover, TNBC patients identified by the WNT/β-catenin network classifier have a greater risk of metastases [4]. The complexity of the disease also stems from the lack of accepted predictors of response to therapy. More importantly, as TNBC lacks oestrogen, progesterone, and HER2 receptors, it is unresponsive to currently available targeted approaches [5]. Thus, new prognostic molecular markers and therapeutic target(s) are required. Currently, many clinical trials targeting specific receptors in tumour and stromal cells are ongoing, and targeted therapies based on immunohistochemical studies are under investigation [6,7].

The human IL-3R is a heterodimeric receptor consisting of an IL-3 specific binding subunit, the α chain (IL-3Rα) [8], and a common β chain (βc) [9] shared with the granulocyte–macrophage colony-stimulating factor (GM-CSF) and interleukin-5 (IL-5) receptor [10,11]. Hemopoietic stem cell commitment as well as monocyte [12], eosinophil [13], basophil [14,15], and neutrophil [16] functional activation requires IL-3R. IL-3 binding to its receptor (IL-3R) triggers a variety of cellular signals upholding the homeostasis of the haemopoietic tissue [17,18]. However, vascular cells also express IL-3R, the activation of which increases the expression of adhesion molecules as well as the inflammatory and immune responses [19,20,21]. We have recently shown that the interleukin-3 binding subunit (IL-3Rα) is expressed in human TNBC samples by endothelial and inflammatory cells [22]. IL-3 was originally described as a haematopoietic factor [17]; however, IL-3, mainly released by tumour-infiltrating lymphocytes (TILs) [23], acts as a proinflammatory and proangiogenic cytokine [24]. Moreover, it has been shown that, in response to IL-3, tumour-derived endothelial cells (TECs) release extracellular vesicles (EVs) able to promote vessel growth [25], epithelial-to-mesenchymal transition (EMT), TNBC metastatic spread [22], and tumour immune evasion by upregulating the programmed cell death-ligand 1 (PD-L1) in myeloid cells [26]. All these data sustain the contribution of IL-3 signalling in the tumour microenvironment (TME). The TME consists of different cell types that provide the ground for tumour development and progression either directly through different mechanisms or by means of the release of soluble factors. Moreover, each of these events most likely contributes to the dynamic changes triggering both tumour immune editing and tumour plasticity [27]. Tumour cell plasticity refers to the dynamic changes that affect cancer cells and explains the reversible mesenchymal transition, the acquisition of stemness traits, and the epithelial-to-endothelial transition [28]. The acquisition of an endothelial phenotype by tumour cells and their ability to form blood vessel lacunae is a process denoted as vascular mimicry (VM), generally referred to as a hallmark of TNBC aggressiveness [29,30,31]. VM represents an alternative mechanism of tumour adaptation to the demands of oxygen and nutrients required for their outgrowth [32] and spreading [33]. Indeed, in TNBC, VM correlates with different prognostic features including haematogenous and lymph node metastases, and the rate of survival [34].

Therefore, the pleiotropic actions of the IL-3/IL-3Rα axis provide the rationale to explore IL-3Rα as a prognostic additional biomarker, and a potential therapeutic target in TNBC aggressiveness and, thus, a forthcoming therapeutic challenge.

## 2. Materials and Methods

### 2.1. Sample Recruitment

To evaluate the clinical significance of the IL-3Rα expression, we selected patients diagnosed with TNBC who underwent surgical treatment and with available follow-up data (*n* = 421). According to the guidelines and clinical indications, some patients were also treated with neo-adjuvant chemotherapy, adjuvant chemotherapy, radiation therapy, or a combination of these treatment modalities. The data of 163 primary TNBC patients who underwent surgery from 2001 to 2019 were collected from the Pathology Unit of Città della Salute e della Scienza Hospital in Turin, while those of 258 patients undergoing surgeries from 2000 to 2015 were recovered from the Department of Histopathology of Oncologic Hospital in Cagliari, Italy. The specimens were processed as previously described [22] in the two different Pathology Units. Three experienced pathologists independently evaluated all tumour cases. Histologic subtyping referred to the 2019 WHO classification [35]. The TNBC database includes the personal and medical data collected from the medical records of each TNBC patient. The study was conducted in accordance with the guidelines and regulations defined by the Research Ethics Committee for Human Biospecimen Utilisation (Department of Medical Sciences—ChBU) of the University of Turin and from the Research Ethics Committee of Sardinia Region (#224/CE/12). Written consent was not required considering the retrospective nature of the study and no impact on patients’ care. Tumour samples were analysed using IHC to assess the expression of IL-3Rα (anti-IL-3Rα clone S-12 antibody Santa Cruz Biotechnology, Dallas, TX, USA, sc-455). To classify each individual tumour sample as IL-3Rα-positive or -negative, the results of IHC were compared via light microscopy using acute myeloid leukaemia cells (M07 cell line) as a positive control, which displays a high receptor expression. Based on this control, we categorised tumours as IL-3Rα-negative or -positive (cut-off 1-3 plus).

### 2.2. Immunohistological Analysis

Tissue microarrays including representative formalin-fixed, paraffin-embedded (FFPE) blocks of TNBC specimens were used to cut 3 μm thick tissue sections for haematoxylin-and-eosin stains (H&E). The TIL evaluation was reported as the percentage of cells in the stromal tissue inside the invasive tumour, avoiding areas with crushed artefacts, necrosis, and inflammation around biopsy sites or hyalinisation [36]. Immunohistochemistry for the expression of IL-3Rα was performed as previously described [22].

### 2.3. Cell Lines

The following TNBC cell lines were used: MDA-MB-231, MDA-MB-453, MDA-MB-436, MDA-MB-157, BT-549, HCC-1395, and Hs-578T. The pathological features of each cell line are reported in Appendix A; the MCF10A cell line, a non-neoplastic breast cancer cell subtype, was cultured as indicated by the manufacturer and served as a negative control. All cell lines were provided by the ATCC (Manassas, VA, USA) and cultured as indicated by the manufacturers. The M07 leukaemic cells were established in the lab [37]. Appendix A reports the details.

### 2.4. Western Blot Analysis

Cells were lysed using a RIPA buffer and processed as previously described [22]. The following antibodies were used: anti-Vimentin (Abcam #ab8978), anti-N-Cadherin (Abcam #ab18203), anti-GAPDH (Abcam # ab8245), anti-β-Actin (Abcam #ab8227), anti-CD31 (Abcam #ab28364), and anti-IL-3Rα/CD123 (R&D Systems #MAB301-100). Secondary antibodies conjugated with peroxidase were purchased from Cell Signalling Technologies (Danvers, MA, USA). Appendix A reports the details.

### 2.5. FACS Analysis

For the FACS analysis of IL-3Rα/CD123 surface marker, MDA-MB-231, MDA-MB-453, MDA-MB-436, MDA-MB-157, BT-549, Hs-578T, HCC-1395, MCF10A, and M07 cells were harvested and stained with human anti-CD123 antibody for 30 min. Flow cytometric analysis was performed using a Cytoflex flow cytometer (Beckman Coulter, Brea, CA, USA). Appendix A reports the details.

### 2.6. Real-Time PCR Analysis

Real-time polymerase chain reaction (PCR) was performed to detect SLUG, ZEB 1, and TWIST transcript in Hs-578T, HCC-1395, MDA-MB-231, and MDA-MB-436 cell lines untreated or treated with IL-3 (5 ng/mL of recombinant human IL-3) (BD Biosciences, San Jose, CA, USA) for 24 h. The Primer sequences and additional details are reported in Appendix A.

### 2.7. Tube-like Structure Formation Assay

Cells were non-enzymatically detached and seeded onto a thick layer of growth factor reduced Matrigel (BD Biosciences, San Jose, CA, USA) in a 24-well plate (Corning, Corning, NY, USA). Appendix A reports the details.

### 2.8. In Vivo Model

Animal studies were performed in accordance with the Italian National Institute of Health Guide for the Care and Use of Laboratory Animals. All procedures were approved by the Ethics Committee of the University of Turin and the Italian Health Ministry (protocol No. 833/2020-PR). The Committee approved the research before the study began. Mice were housed according to the guidelines of the Federation of European Laboratory Animal Science Association. Blinded investigators (at least 2) assessed the outcome. Tumours were obtained by injecting MDA-MB-231 or MDA-MB-436 cells (2 × 10^6^ cells) in 100 µL of growth factor reduced Matrigel into the mammary fat pad of NSG mice purchased from Charles River Laboratories (Calco, Italy) (8 weeks/female) (3 mice/group: control vs. rhIL-3-treated mice (20 ng/mL)). From T0, corresponding to tumour cell injection, the animals were locally treated with saline or rhIL-3 (20 ng/mL) every 2 days for 4 weeks. Once a week, the size of the tumours was blindly evaluated using a digital calliper: Maximum and minimum diameters were measured, and the volume was calculated using the following ellipsoid formula: (4/3π (d/2)^2∗D/2). Mice were euthanised using a CO_2_ chamber 4 weeks later. Primary tumours were collected and fixed as previously described [22], for further analyses. Immunohistochemistry was performed using an automated slide-processing platform (Ventana BenchMark AutoStainer, Ventana Medical Systems, Tucson, AZ, USA), with Universal DAB Detection Kit detection systems. Briefly, 5 µm paraffin-embedded tumour sections were stained with CD31 and PAS to quantify CD31+ vessels and VM (expressed as CD31−/PAS+ vessels). Ten sections/tumour were examined using ImageJ 1.50g software (Wayne Rasband, National Institutes of Health, Bethesda, MD, USA), and the results are expressed as the number of CD31+/PAS+/fields ± SEM. PD-L1 expression was evaluated using an anti-human PD-L1 rabbit polyclonal antibody (Abcam #ab233482) diluted in a ratio of 1:100. Secondary HRP-labelled anti-rabbit antibody (Goat Anti-Rabbit IgG (H + L)-HRP Conjugate, Bio-Rad, Hercules, CA, USA) was used at 1:1000 dilution for 1 h at room temperature. Sections were counterstained with haematoxylin, dehydrated, and mounted. The quantification of the PD-L1 expression was performed using Fiji software (ImageJ). Lung metastases were counted according to human PD-L1 expression. The results are expressed as the mean ± SEM of PD-L1-positive area (related units) per sample (*n* = 3/each condition).

### 2.9. Bioinformatics Analysis

Using The Cancer Genome Atlas (TCGA), limited to the BRCA data, the datasets were downloaded from the TCGA website (https//tcga-data.nci.nih.gov, accessed on 14 November 2021) for gene expression analyses [38,39,40]. Appendix A reports the details.

### 2.10. Sample Size Calculation

For animal studies, according to our previous data, the minimum sample size that would permit us to detect a 40% difference between the experimental groups with 90% power was *n* = 3 mice/group. For patient validation, based on the data provided by Urru et al. [41], a simulation study of the number of events per variable in logistic regression analysis [42] revealed that 250 patients are needed to test the independent predictive power of the IL-3Rα expression.

### 2.11. Statistical Analysis

Data are representative of at least 3 independent in vitro experiments, performed in triplicate. Comparison between 2 groups was performed using Student’s *t*-test, while comparisons between 3 or more groups were performed using a one-way ANOVA test, and the significance level (*p* < 0.05, *p* < 0.01, *p* < 0.001) was evaluated using the Newman–Keuls multicomparison post hoc test. The correlation and distribution of IL-3Rα and baseline clinical–pathological parameters were assessed using the χ^2^ test and Fisher’s exact test. Baseline parameters such as age at diagnosis (we arbitrarily set the cut-off at 55 years old since we failed to detect differences in the median of age); histotype (ductal, lobular, or others); pT = primary tumour; pN = lymph nodal invasion; M = metastasis (TNM classification); G = grade (G1, G2, G3); Ki67 (%), and the percentage of TILs = tumour-infiltrating lymphocytes in every single tumour specimen were considered. The overall survival (OS) analysis was performed using the Kaplan–Meier estimates. We used Cox regression and the Wilcoxon–Breslow–Gehan test to assess the hazard ratio and expected-versus-observed events in the two groups of patients. Cox multivariate adjustment was performed including the IL-3Rα expression and pT. The association between the relative gene expression values was performed through Spearman’s correlation analysis. Logistic regressions were performed to evaluate the synergistic effect of several genes in discriminating the tumour subtype based on its aggressiveness. The receiver operative characteristic (ROC) curve considering the area under the curve (AUC) was used to judge the discrimination potential of the models. Statistical analyses were performed using R v4.1.2, R Core Team (2017); R: a language and environment for statistical computing; R Foundation for Statistical Computing, Vienna, Austria. URL (https://www.R-project.org/, accessed on 1 November 2021); Software and Bioconductor (https://www.r-project.org/, accessed on 1 November 2021). Figure results were plotted using the Ggplot2 R package (https://CRAN.R-project.org/package=ggplot2, accessed on 25 January 2022) and STATA (https://www.stata.com, accessed on 20 April 2021), Copyright 1985–2019 StataCorp LLC, serial number: 401709301228; licensed to Università degli Studi di Torino). Statistical significance was set at *p* < 0.05.

## 3. Results

### 3.1. IL-3Rα Is Expressed and Associated with Poor Prognosis in Human TNBC

We performed a retrospective study to evaluate the expression of IL-3Rα in TNBC samples obtained from two different centres in Italy (*n* = 421). Notably, IL-3Rα expression was found in 291 out of 421 human TNBC samples, and at baseline, IL-3Rα-positive patients displayed a significantly high nodal invasion (N2) (*p* = 0.026) (Table 1). The representative images of positive and negative IL-3Rα samples are shown in Figure 1A,B. In our cohort, we detected 133 events, 99 in IL-3Rα-positive and 34 in IL-3Rα-negative patients. The hazard ratio for IL-3Rα-positive samples corresponded to 1.50 (CI = 95% 1.01–2.2; *p* = 0.04) and was near to significance after multivariate adjustment (hazard ratio = 2.01:0.89–4.89, *p* = 0.089). Additionally, using the Wilcoxon–Breslow–Gehan test, we found that the expected events were significantly lower (IL-3Rα-positive: 88 vs. 99) and higher (IL-3Rα-negative: 45 vs. 34) than the observed (*p* = 0.0245) (Figure 2).

### 3.2. IL-3Rα Activation Impacts EMT and Reprograms TNBC Cells towards an Endothelial-like-Phenotype

To gain insights into the biological relevance of IL-3Rα in TNBC, its expression was first investigated by selecting a panel of TNBC cell lines displaying different phenotypic features (Appendix A). As in human samples, IL-3Rα was expressed in TNBC cell lines (Figure 3A,B).

Based on our retrospective study, we selected four different TNBC cell lines (two originated from pleural effusion and two from primary tumours) to evaluate whether the activation of the IL-3/IL-3Rα axis impacts TNBC aggressiveness. For this purpose, IL-3 was used to activate IL-3Rα and to investigate cell proliferation and the expression of EMT markers in TNBC. IL-3 was unable to promote TNBC proliferation in all the analysed cell lines. Therefore, Vimentin and N-Cadherin were selected as EMT markers (Figure 4A–D). IL-3 upregulated N-Cadherin in MDA-MB-231 and MDA-MB-436 and Vimentin in Hs-578T and HCC-1395 cells, while downregulated N-Cadherin in HCC-1395 cells (Figure 4D). β-catenin expression was not affected by IL-3. The SLUG, TWIST, and ZEB1 mRNA expression levels were analysed and are reported in Figure 4E–H. Possibly due to the high basal expression of these genes in TNBC cell lines, IL-3 significantly upregulated their expression only in HCC-1395 cells (Figure 4H).

Epithelial-to-endothelial transition and the formation of tumour-cell-derived vessel-like structures (vascular mimicry (VM)) are common hallmarks of TNBC aggressiveness [32,34]. Thereby, the ability of IL-3 to reprogram TNBC cells towards an endothelial-like phenotype was investigated in vitro using a three-dimensional tube-formation assay. As shown in Figure 5A,B, IL-3 induced morphological changes in cancer cells, translating in their ability to form tube-like structures.

### 3.3. IL-3 Boosts VM and PD-L1 Expression in Primary Tumours and Lung Metastases

The impact of the IL-3/IL-3Rα axis in TNBC biology was also investigated in vivo by injecting MDA-MB-231 and MDA-MB-436 cells in NSG mice. These cell lines were selected for their parallel EMT marker modulation, phenotype, and growth in mice [43]. The protocol is shown in Figure 6A. We failed to detect significant differences in primary tumour size between the untreated and IL-3-treated animals (Appendix A). Therefore, based on the in vitro results, the tumour-associated vasculature was examined. As shown in Figure 6B–D and Figure 6F–H, tumours from animals treated with IL-3 displayed a significant increase in PAS-positive/CD31-negative vessels, corresponding to the vascular network built by tumour cells and denoted as VM. Conversely, vessels, mainly formed by endothelial cells, were found in control mice. Moreover, a significant increase in vessel size was detected in the animals treated with IL-3 (Figure 6E,I).

We have recently shown that the EVs released by TEC upon IL-3Rα blockade modulate PD-L1 expression in myeloid cells [26]. Since cancer cells also express PD-L1, and its expression contributes to TNBC aggressiveness, PD-L1 was also evaluated in our in vivo model. IL-3 was able to significantly increase PD-L1 expression in primary tumours (Figure 7A,B), as well as the number of lung lesions and the magnitude of the scattered PD-L1-positive tumour cells in the lung (Figure 7C–H). Overall, these data provide evidence that the IL-3/IL-3Rα signalling is a crucial remodelling pathway contributing to TNBC aggressiveness.

### 3.4. Bioinformatics Analysis on TCGA Data Recognises IL-3Rα as a Marker of TNBC Aggressiveness

The association between the expression of IL-3Rα and breast cancer aggressiveness was also evaluated by interrogating The Cancer Genome Atlas (TCGA) data. The results of the differential expression analysis, based on a limited number of TNBC data, indicated that IL-3Rα was highly expressed in TNBCs compared with luminal breast cancers (*p* = 0.017, *p*adj = 0.026, (Figure 8A). The TCGA data analysis also identified EMT and VM coding genes as differentially expressed in the two breast tumour groups. Moreover, IL-3Rα positively correlated with VIM (rho = 0.52, *p* < 2.2 × 10^−16^), ZEB2 (rho = 0.47, *p* < 2.2 × 10^−16^), SPRY2 (rho = 0.42, *p* < 2.2 × 10^−16^), MMP2 (rho = 0.37, *p* < 2.2 × 10^−16^), and CD274 (PD-L1) (rho = 0.38, *p* < 2.2 × 10^−16^) (Figure 8B). Univariate and multivariate logistic regression analyses were performed by selecting those genes with the best individual *p*-value (SNAI1, NFKB2, ZEB1, SMAD3, VIM, CD274, CTNNB1, MMP2, SPRY2, SNAI2, and IL-3Rα) as candidate predictors for breast cancer aggressiveness (Figure 8B). Finally, by considering the genes recapitulating IL-3 biological functions, we proposed a model, including IL-3Rα, SNAI1, ZEB1, VIM, CTNNB1, MMP2, SPRY2, and CD274, that remarkably discriminates cancer aggressiveness (AUC = 0.86 95% CI = 0.82–0.89) (Figure 8C).

## 4. Discussion

In the present study, we identified IL-3Rα as an additional TNBC prognostic biomarker and provided the proof of concept that TNBC aggressiveness also stems from the IL-3/IL-3Rα signalling pathway.

TNBC is a deep heterogeneous breast cancer lacking canonical breast cancer markers such as the oestrogen, progesterone, and HER2 receptors [1], and thus targeted treatments [5]. Herein, we demonstrated that 291 out of the 421 TNBC specimens examined (69%) were positive for IL-3Rα. The TCGA data analysis, grouping 1080 luminal breast cancer samples and 125 TNBC samples, confirmed this observation.

IL-3 is a proinflammatory and proangiogenic cytokine mainly released by activated T lymphocytes and mast cells in the TME [44] and acts as an autocrine factor for human breast and kidney tumour-derived endothelial cells [24]. The biological relevance of IL-3 and its receptor has been extensively investigated in the haemopoietic and vascular compartments [45], while their role in TNBC is yet unknown. We herein demonstrated that the expression of IL-3Rα in human TNBC samples correlates with the presence of metastatic nodes and with a poor overall survival rate. The data from TCGA established that several genes closely linked to EMT, and angiogenesis positively correlate with the IL-3Rα expression in TNBC. The angiogenic process is instrumental for tumour development and metastatic spread [46]. However, cancer dissemination also relies on the ability of cancer cells to build their own vascular network embedded into the endothelial layer, in a process known as VM [47]. Notably, EMT regulators and EMT-related transcription factors are generally regarded as hallmarks of VM and TNBC progression [48]. Consistent with the TCGA molecular annotation, our data proved that the IL-3/IL-3Rα axis drives changes in EMT markers and acts as a booster of VM in vivo, translating into tumour cell homing and outgrowth in the lung.

Tumour cells sense cues released by TME components and promptly adapt to any favouring challenge [49]. These reciprocal interactions within the TME translate into tumour cell reprogramming and shift towards specific “phenotypic identity”, outgrowth, and dissemination [50]. Among these reprogramming plans, the dynamic and reversible EMT to mesenchymal–epithelial transition and the epithelial-to-endothelial switch appear relevant [50]. We herein reported that the activation of IL-3Rα drives endothelial-like features, enhances the formation of large blood lacunae, which replace and fulfil endothelial cell functions, and boosts TNBC metastatic spread. VM is highly prevalent and more frequently identified in TNBC compared with luminal or HER2+ breast cancers and is generally associated with tumour progression [51,52]. Therefore, the close relationship between VM, EMT, and the expression of a functional IL-3Rα provides additional insight into the role of IL-3/IL-3Rα in the complex molecular programme guiding TNBC metastatic spread and aggressiveness. Moreover, since resistance to antiangiogenic treatments also relies on the occurrence of VM [53], the activation of the IL-3/IL-3Rα-mediated signalling may represent one of the rescue pathways granting tumour cell survival in this clinical scenario. In addition, consistent with “omics technologies” [54], our results provide evidence of the heterogeneity of IL-3Rα response in different TNBCs (the ability to induce the expression of EMT marker was tumour-specific), as well as on the ability of IL-3Rα to activate common pathways (the induction of the epithelial-to-endothelial switch was detected in all the cell lines tested).

The TME in TNBCs is unique and dynamically contributes to tumour immune evasion via discrete mechanisms including the aberrant expression of immune checkpoint proteins, such as PD-L1 and the programmed cell death-ligand 1 receptor [55]. PD-L1 is highly expressed in several cancer types [56] and much more in TNBCs compared with other breast cancer subtypes [57]. Genetic and epigenetic mechanisms, as well as inflammatory stimuli, control PD-L1 expression in cancers [58,59]. We demonstrated that IL-3Rα activation increases PD-L1 expression in primary tumours and in metastatic lung lesions, sustaining a role in the control of anti-tumour immune response. Soluble factors released in the TME also dictate the clinical benefits of checkpoint inhibitors [54]. As an example, the vascular endothelial growth factor (VEGF) was found instrumental for TME immunosuppression by inducing vascular defects and rearranging the anti-tumour immune response [60]. Therefore, our observations that, in tumour-bearing mice, the IL-3Rα-mediated signal blunted the “physiological” angiogenesis, boosted the VM, and increased the PD-L1 expression indicate that, as VEGF, IL-3 can “highjack” the TME, thereby contributing to tumour evasion.

## 5. Conclusions

A molecular classification based on both transcriptomics and genomics is mandatory to identify novel targets and develop innovative therapeutic options [61]. Our study, besides refining the TNBC molecular annotation, proposed a model recapitulating cancer aggressiveness. Moreover, since tumour molecular profiling supported the IL-3Rα expression in human samples and its potential role in driving tumour aggressiveness in vivo, we proved that its activation drives tumour progression; thus, investigating IL-3Rα as a potential therapeutic target presents a valuable challenge.

## Figures and Tables

**Figure 1 cancers-14-03918-f001:**
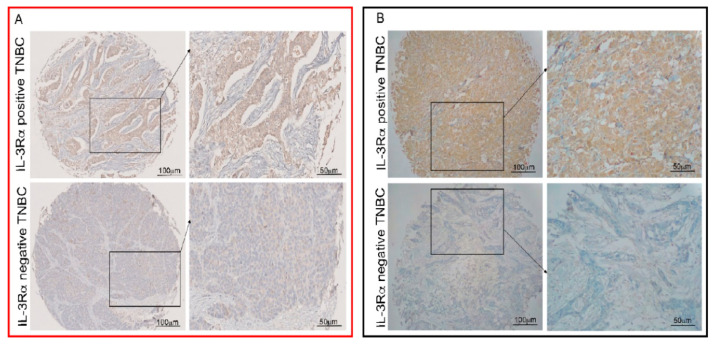
Human TNBC cells express IL-3Rα. Representative tissue microarrays from human TNBC samples were stained with anti-IL-3Rα antibodies. IL-3Rα-positive and -negative TNBC samples are shown. Original magnification 10× and 20×, scale bar: 100 µm and 50 µm, respectively. Panel (**A**) (red square) refers to Turin samples and panel (**B**) (black square) to Sassari samples.

**Figure 2 cancers-14-03918-f002:**
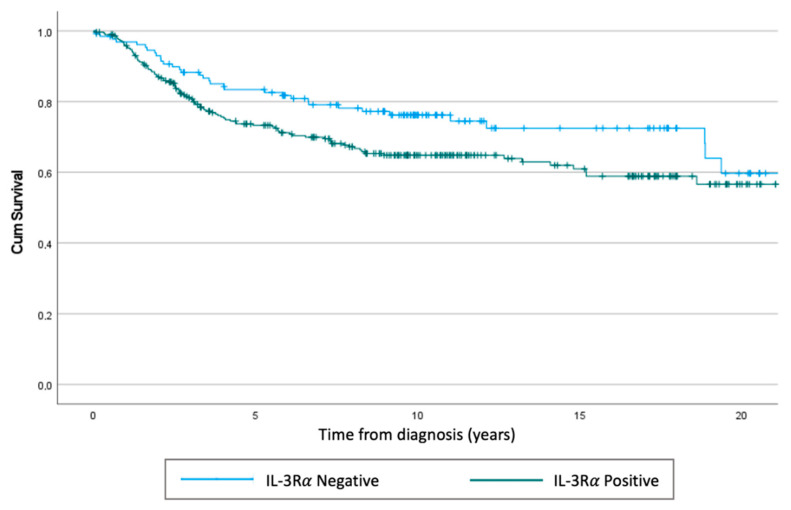
Kaplan–Meier of overall survival. Using the Cox regression test, we obtained a *p* = 0.04, while using the Wilcoxon–Breslow–Gehan test, the *p* corresponded to *p* = 0.0245.

**Figure 3 cancers-14-03918-f003:**
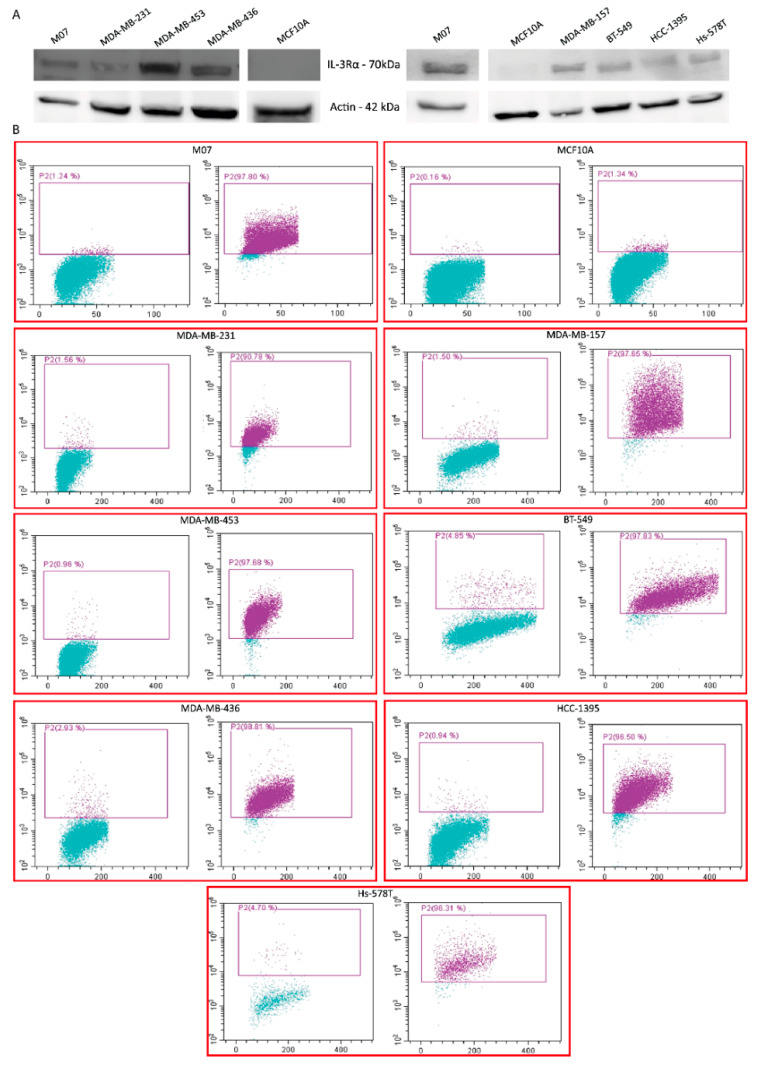
IL-3Rα expression in different TNBC cell lines: (**A**) representative Western blot of IL-3Rα expression in the indicated TNBC cell lines. Actin served as housekeeping gene. The uncropped blots are shown in Appendix A; (**B**) FACS dot plots of IL-3Rα expression (purple dots) and appropriate IgG controls (left panel in each cell line) of the indicated TNBC cell lines. M07 and MCF10A cells served as positive and negative controls, respectively.

**Figure 4 cancers-14-03918-f004:**
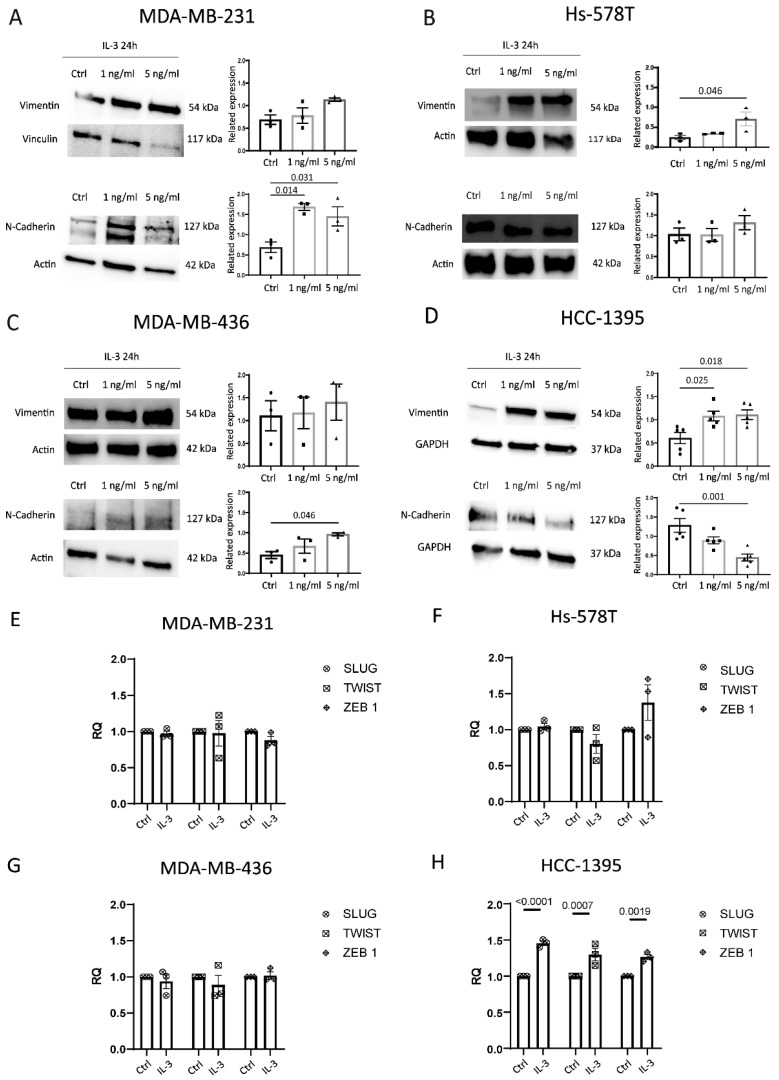
IL-3Rα activation impacts EMT: (**A**–**D**) representative Western blot and quantification of Vimentin and N-Cadherin expression levels in MDA-MB-231, Hs-578T, MDA-MB-436, and HCC-1395 cells untreated (Ctrl, circles on diagrams) or treated with IL-3 (1 ng/mL, squares on diagrams or 5 ng/mL, triangles on diagrams). Data are expressed as the mean ± SEM normalised to housekeeping genes (Vinculin, Actin, and GAPDH). (**E**–**H**) SLUG, TWIST1, and ZEB 1 mRNA expression levels in MDA-MB-231, Hs-578T, MDA-MB-436, and HCC-1395 cells untreated (Ctrl) or treated with IL-3 (1 ng/mL or 5 ng/mL). Data are expressed as the mean ± SEM normalised to housekeeping genes. The uncropped blots are shown in Appendix A.

**Figure 5 cancers-14-03918-f005:**
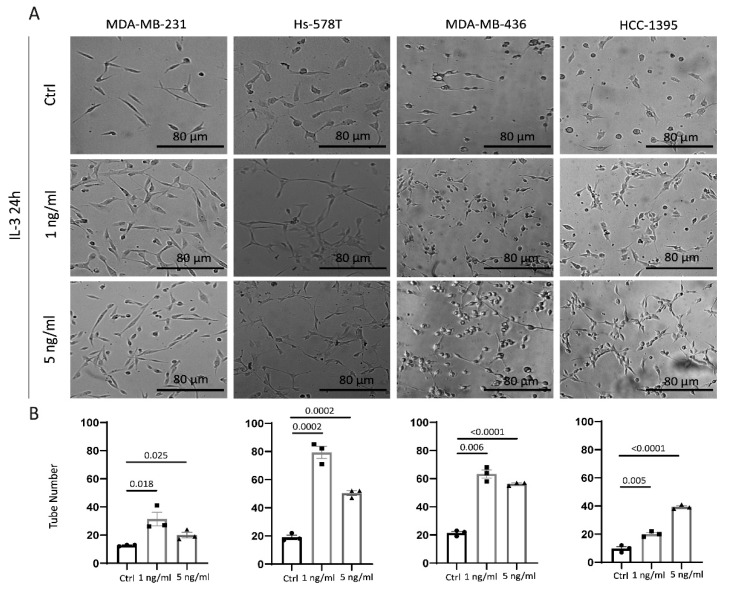
IL-3Rα activation reprograms TNBC cells towards an endothelial-like phenotype: (**A**) phase images of cells plated on growth factor reduced Matrigel in serum-free medium or supplemented with 1 ng/mL or 5 ng/mL of IL-3; (**B**) quantification of the vessel number was performed over at least three independent replicates for each cell line. MDA-MB-231, Hs-578T, MDA-MB-436, and HCC-1395 cells untreated (Ctrl, circles on diagrams) or treated with IL-3 (1 ng/mL, squares on diagrams or 5 ng/mL, triangles on diagrams) are shown. The results are expressed as mean ± SEM. Original magnification 20×, scale bar: 80 µm. Comparisons were performed using one-way ANOVA followed by Tukey’s multiple-comparison test.

**Figure 6 cancers-14-03918-f006:**
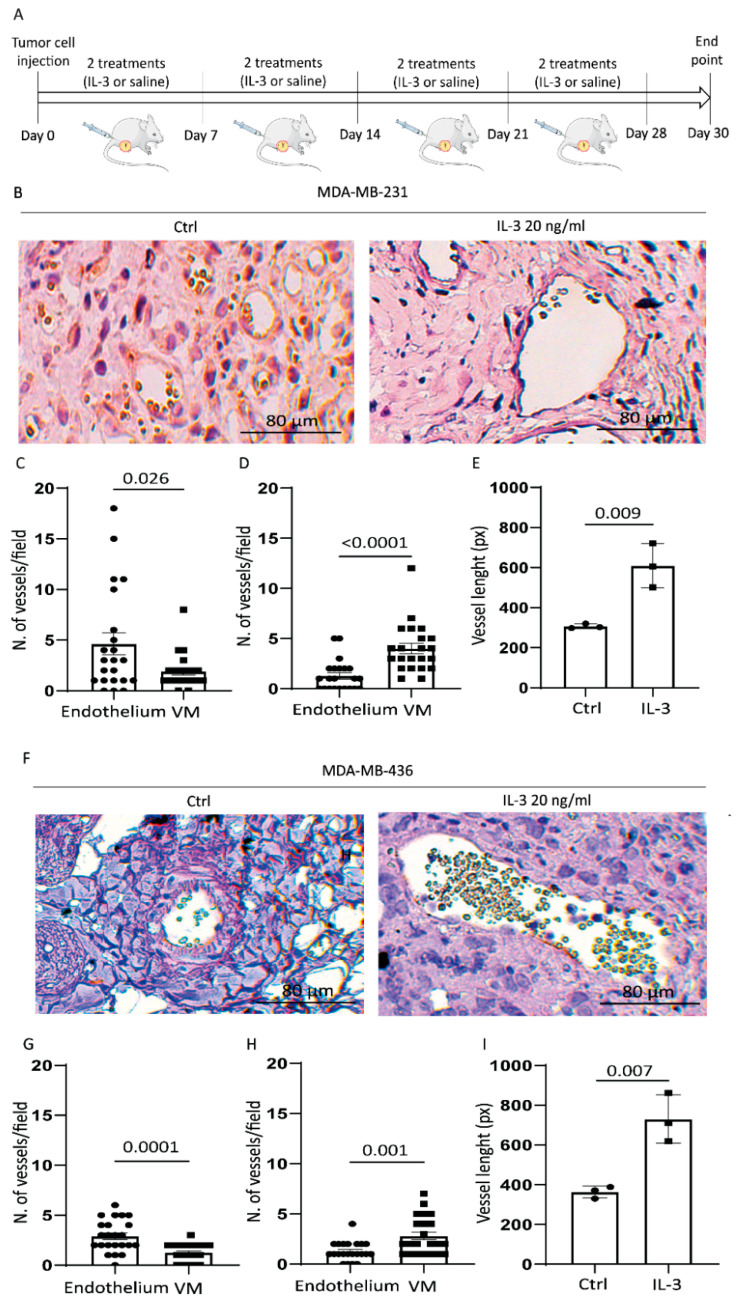
IL-3Rα activation promotes VM and increases the vessel size in vivo: (**A**) NSG mice were injected with tumour cells (as indicated) and treated from day 0 with saline or IL-3 (20 ng/mL) twice a week for 4 weeks. The figure was partly generated using Servier Medical Art templates, which are licensed under a Creative Commons Attribution 3.0 Unported License; https://smart.servier.com, (accessed on 5 August 2022). Tumours were recovered at day 30; (**B**) representative images of MDA-MB-23- derived tumours untreated (Ctrl) or treated with IL-3 (20 ng/mL) stained with anti-CD31 antibody (brown colour) and PAS (violet colour); (**C**,**D**) relative quantification of VM/endothelium expressed as the number of CD31+/PAS+ per field ± SEM ((**C**) corresponds to control, while (**D**) refers to IL-3 treatment); (**E**) quantification of the vessel size in MDA-MB-231-derived tumours. The results are expressed as mean ± SEM; (**F**) representative images of MDA-MB-436-derived tumours untreated (Ctrl) or treated with IL-3 (20 ng/mL) stained with anti-CD31 antibody and PAS; (**G**,**H**) relative quantification of VM/endothelium expressed as the number of CD31+/PAS+ per field ± SEM ((**G**) corresponds to control, while (**H**) refers to IL-3 treatment); (**I**) Quantification of the vessel size in MDA-MB-436 derived tumours. Circles on diagrams represent the quantity of CD31+ structures per field, squares represent the quantity of PAS+ structures per field. The results are the mean ± SEM. Original magnification 20×, scale bar: 80 µm. Comparisons were performed using Student’s *t*-test.

**Figure 7 cancers-14-03918-f007:**
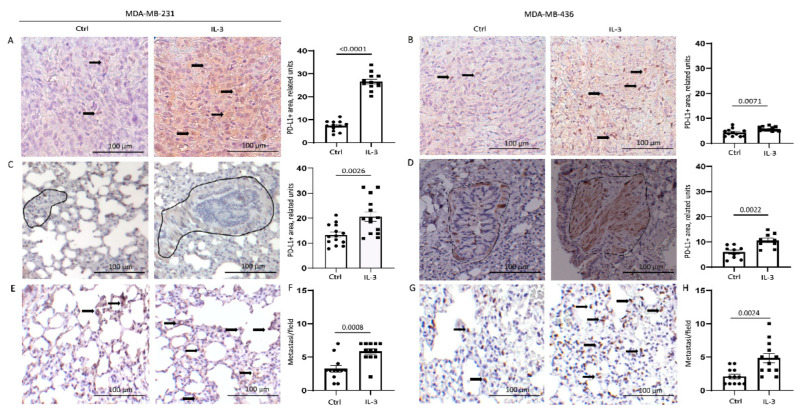
IL-3Rα activation boosts PD-L1 expression in primary tumours and lung metastases: (**A**,**B**) representative images and quantification of PD-L1 expression in primary tumours derived from mice injected with MDA-MB-231 (**A**) and MDA-MB-436 (**B**) cells untreated or treated with IL-3; (**C**,**D**) representative images and quantification of PD-L1 expression in the lung tissues of mice injected with MDA-MB-231 (**C**) and MDA-MB-436 (**D**) cells left untreated or treated with IL-3. Diagram data are presented as PD-L1+ area/lung ± SEM; (**E**,**G**) representative images of scattered PD-L1+ cells in the lung tissues of mice injected with MDA-MB-231 (**E**) and MDA-MB-436 (**G**) cells untreated or treated with IL-3; (**F**,**H**) diagrams reporting the number of lung metastases counted according to PD-L1 expression in mice injected with MDA-MB-231 (**F**) or MDA-MB-436 (**H**) cells untreated or treated with IL-3. Diagram data are expressed as PD-L1+ metastasis/field ± SEM. Circles on diagrams represent PD-L1 quantification in control mice tissues, squares represent the quantity of PD-L1+ structures in the mice treated with IL-3. Original magnification 20× scale bar: 100 µm. Comparisons were performed using Student’s *t*-test.

**Figure 8 cancers-14-03918-f008:**
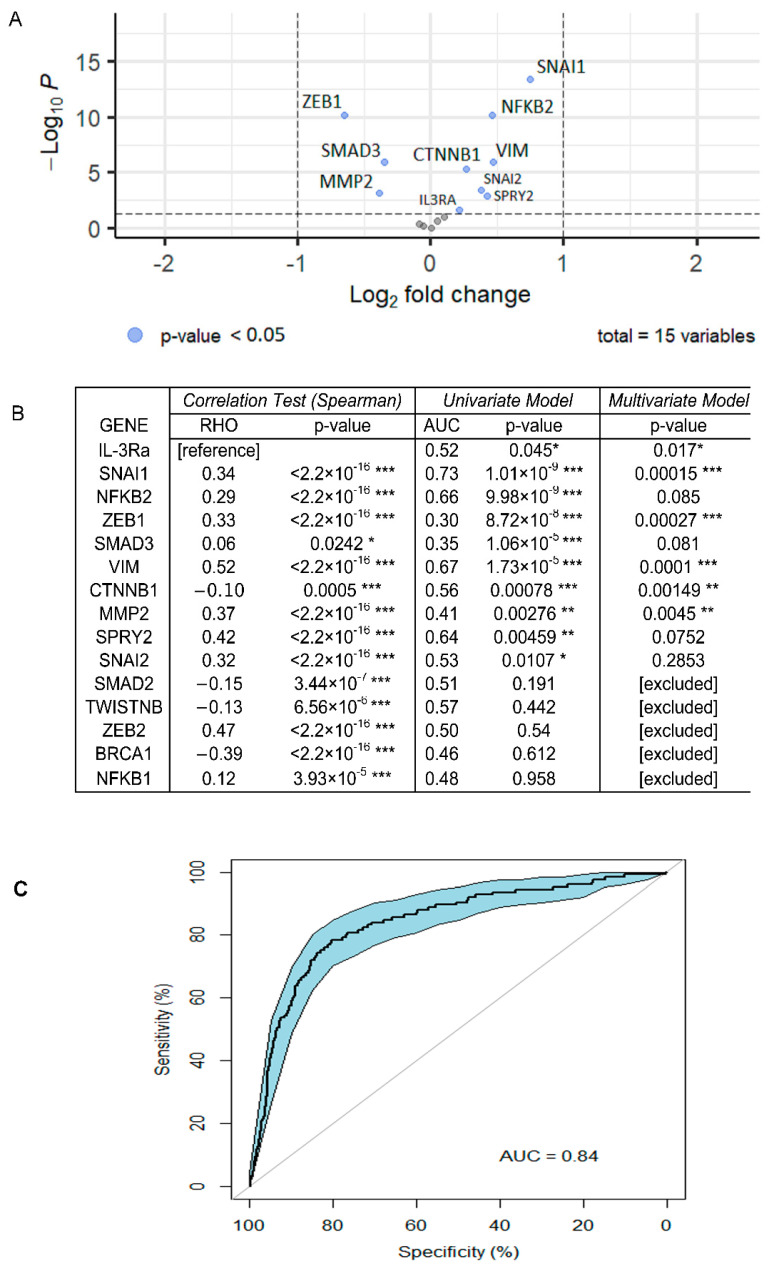
Bioinformatics analysis of TCGA dataset: (**A**) volcano plots of DEGs in TNBC and luminal breast cancers. Log2FC is displayed on the y-axis and the −log10 of *p*-value on the x-axis. Dots represent upregulated and downregulated genes; the blue points are those whose *p*-values were statistically significant (*p* < 0.05); (**B**) correlation of gene expression between IL-3Rα and genes involved in EMT and CD274 (PD-L1) (correlation test). The closer the RHO corresponds to 1, the more the correlation. Results of the univariate model show the discriminating power of single genes (AUC). The *p*-value of the multivariate model showed that all selected genes were relevant for the efficiency of the model. * *p*-value ≤ 0.05; ** *p*-value ≤ 0.01; *** *p*-value ≤ 0.001; (**C**) ROC curve representing the multivariate logistic model. This model, composed of IL-3Rα, SNAI1, ZEB1, VIM, CTNNB1, MMP2, SPRY2, and CD274 (PD-L1), showed the effectiveness in discriminating breast cancer aggressiveness. (AUC = 0.86 95% CI = 0.82–0.89).

**Table 1 cancers-14-03918-t001:** Clinical and pathological features of TNBC patients. Histologic classification refers to the WHO classification. IL-3Rα = IL-3 receptor α, NST = no special type, N/A = not available, pT = primary tumour, pN = lymph nodal invasion, M = metastasis, G = grade, TILs = tumour-infiltrating lymphocytes. * The *p*-value refers to differences between IL-3Rα expression (positive and negative) using Fisher’s exact test (in bold the significant data).

	Total	IL-3Rα-Positive	IL-3Rα-Negative	*p*-Value
	*n* = 421	*n* = 291 (69%)	*n* = 130 (31%)	
**Age, *n* (%)**				0.394
<55 y	191 (45%)	128 (44%)	63 (48%)	
≥55 y	230 (55%)	163 (56%)	67 (52%)	
**Histologic subtype, *n* (%)**				0.169
NST	295 (70%)	201 (69%)	94 (72%)	
Lobular	20 (5%)	18 (6%)	2 (2%)	
Others	96 (23%)	64 (22%)	32 (24%)	
N/A	10 (2%)	8 (3%)	2 (2%)	
**Primary Tumour, *n* (%)**				0.794
pT1	162 (38%)	117 (40%)	45 (35%)	
pT2	194 (46%)	130 (45%)	64 (49%)	
pT3	31 (8%)	22 (8%)	9 (7%)	
pT4	24 (6%)	15 (5%)	9 (7%)	
N/A	10 (2%)	7 (2%)	3 (2%)	
**Lymph node involvement, *n* (%)**				
pN0	230 (55%)	157 (54%)	73 (56%)	
pN1	94 (22%)	61 (21%)	33 (25%)	
pN2	**40 (10%)**	**36 (12%)**	**4 (3%)**	**0.026** *
pN3	31 (7%)	21 (7%)	10 (8%)	
N/A	26 (6%)	16 (6%)	10 (8%)	
**Metastasis, *n* (%)**				0.899
M0	409 (97%)	283 (97%)	126 (97%)	
M1	5 (1%)	3 (1%)	2 (2%)	
N/A	7 (2%)	5 (2%)	2 (2%)	
**Grade, *n* (%)**				0.115
G1	4 (1%)	3 (1%)	1 (1%)	
G2	56 (13%)	45 (15%)	11 (8%)	
G3	354 (84%)	240 (82%)	114 (88%)	
N/A	7 (2%)	3 (1%)	4 (3%)	
**Ki67, *n* (%)**				0.463
<15%	21 (5%)	13 (4%)	8 (6%)	
≥15%	400 (95%)	278 (96%)	122 (94%)	
**TILs, *n* (%)**				0.204
Present	164 (40%)	106 (37%)	58 (46%)	
Absent	121 (30%)	90 (32%)	31 (25%)	
N/A	124 (30%)	87 (31%)	37 (29%)	

## Data Availability

The data can be shared up on request.

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
