# Peer review of "Interleukin-3-Receptor-α in Triple-Negative Breast Cancer (TNBC): An Additional Novel Biomarker of TNBC Aggressiveness and a Therapeutic Target"

_cancers, 2022, doi:10.3390/cancers14163918_

Round 1

Reviewer 1 Report

The authors present an interesting study identifying IL-3α as a marker for triple negative breast cancer and characterize its role in contributing to the tumor aggressiveness by facilitating vascular remodelling and promoting transition to an endothelial phenotype. One strength of the manuscript is the use of clinical samples.

A few suggestions:

Line 276: In this section, it will be interesting to add whether the 291 patients with higher IL-3α expression also had elevated levels of PDL-1?

Second, in these patient samples is there an increased expression of endothelial-phenotype markers?

The authors have also demonstrated the relevance of IL-3α.

Author Response

We thank the Reviewer for His/Her appreciation of our experimental approach and positive comments.

The authors present an interesting study identifying IL-3α as a marker for triple negative breast cancer and characterize its role in contributing to the tumor aggressiveness by facilitating vascular remodelling and promoting transition to an endothelial phenotype. One strength of the manuscript is the use of clinical samples.

A few suggestions:

Line 276: In this section, it will be interesting to add whether the 291 patients with higher IL-3α expression also had elevated levels of PDL-1?

Second, in these patient samples is there an increased expression of endothelial-phenotype markers?

We thank the Reviewer for His/Her suggestions. At this time, we started to recruit patients for a prospective study to provide additional information on IL-3α expression and OS as well as to perform additional molecular analysis and validate PD-L1 expression and markers of vascular mimicry.

Reviewer 2 Report

The manuscript titled “Interleukin-3-Receptor-α in Triple-Negative Breast Cancer: A Novel Biomarker and Therapeutic Target” describes that the authors want to prove IL-3Rα as a novel marker and target which promotes PD-L1 overexpression of TNBC resulted in lung metastases. Patients’ samples, in vivo and in vitro results indicate IL-3Rα associated with TNBC progression. If the authors could provide deep insight study and therapeutic results that would further prove and support IL-3Rα is a promising therapeutic target. The followings are some concerns and comments that have been pointed out that the authors may want to consider.

Major Concerns and Comments:

1. Line 50 Introduction section: This part needs to extend. More background and information are needed.

2.      Lines 65-71: Please remove unnecessary self-citation.

3.   Line 148, and lines 174-176: Only 3 mice for each group are not enough. Provide your previous data if the experiments were performed under the same condition.

4.      Line 228: Please provide higher resolution images.

5.    Line 233 Figure 2: Do authors have any idea why IL-3Rα is positive or negative with a slight difference after 19 years of diagnosis? Might be a 2% difference.

6.   Lines 233-234: a) Please consistent p as italic, and check throughout the manuscript, for example, line 31 and so on; b) Is it possible to increase the resolution of the image?

7.   Line 264 Figure 4: a) Please increase the resolution of the images; b) Please involve at least a brief description of the statistical method in the figure legend; c) Figure 4C, please confirm N-Cadherin and GAPDH from the same membrane or not? Why does the loading control drop too much? The authors stated “at least 3 independent in vitro experiments”. Please provide other GAPDH etc. with better quality.

8.  Line 292 Figure 5, line 314 Figure 6: Please involve at least a brief description of the statistical method in the figure legend;

9.  Lines 306-309: It’s OK to use the previous study to support current research. But the reviewer doesn’t think it’s a good idea that the authors want to conclude the current TNBC study using previous myeloid cells results.

10.  IL-3 is one of the important pro-inflammatory signaling factors and cytokines, there is no surprise that the receptor IL-3Rα is overexpressed in TNBC. The authors provide HIC results that IL-3Rα increased in clinical tissue samples, and TNBC cell lines to suggest IL-3Rα is a therapeutic target. Are there any therapeutic data and deep insight results to show? Please provide.

Minor Concerns and Comments:

1.      Line 25: Why is “particularly in TNBC”? The reviewer thinks “For example” might be more suitable.

2.      Line 82: “About 163” is not a scientific description.

3.      Line 150: Please specify the dose of rhIL-3 for each mouse.

4.      Line 221: Is there any reason the cut-off age is 55 years old?

5.      Line 260 Figure 3: Please increase the resolution of the images.

6.  I’d suggest the authors include molecular weight in the original images.

Author Response

We thank the Reviewer for His/Her appreciation of our experimental approach and positive comments.

The manuscript titled “Interleukin-3-Receptor-α in Triple-Negative Breast Cancer: A Novel Biomarker and Therapeutic Target” describes that the authors want to prove IL-3Rα as a novel marker and target which promotes PD-L1 overexpression of TNBC resulted in lung metastases. Patients’ samples, in vivo and in vitro results indicate IL-3Rα associated with TNBC progression. If the authors could provide deep insight study and therapeutic results that would further prove and support IL-3Rα is a promising therapeutic target. The followings are some concerns and comments that have been pointed out that the authors may want to consider.

Major Concerns and Comments:

1.Line 50 Introduction section: This part needs to extend. More background and information are needed.

As kindly requested by the Reviewer, we have extended the introduction section and added more information about the dynamic changes in the tumour microenvironment (TME).

  1. Lines 65-71: Please remove unnecessary self-citation.

As kindly requested by the Reviewer, we have removed 1 self-citation and add an additional and older citation on this topic.

  1. Line 148, and lines 174-176: Only 3 mice for each group are not enough. Provide your previous data if the experiments were performed under the same condition.

We thank the Reviewer for His/Her comment. Honestly, we have planned to perform an additional group of animals, however since we observed a significate difference using only 3 animals per group we decided to leave off the experiments to refine and reduce the number of animals.

  1. Line 228: Please provide higher resolution images.

As kindly requested by the Reviewer, we provided higher resolution images.

5.Line 233 Figure 2: Do authors have any idea why IL-3Rα is positive or negative with a slight difference after 19 years of diagnosis? Might be a 2% difference.

We thank the Reviewer for His/Her comment. We do not have an explanation, however it can be speculated that patients’ aging may have concurred to the occurrence of death causes different and independent of the IL-3Rα expression.

  1. Lines 233-234: a) Please consistent p as italic, and check throughout the manuscript, for example, line 31 and so on; b) Is it possible to increase the resolution of the image?

As kindly requested by the Reviewer, the “p” are in italic in the present version of the Ms and we increased the resolution of the images. 

  1. Line 264 Figure 4: a) Please increase the resolution of the images; b) Please involve at leas a brief description of the statistical method in the figure legend; c) Figure 4C, please confirm N-Cadherin and GAPDH from the same membrane or not? Why does the loading control drop too much? The authors stated “at least 3 independent in vitro experiments”. Please provide other GAPDH etc. with better quality.

As kindly requested by the Reviewer, we have i) increased image resolution; ii) added a brief description of the statistical method in the figure legend; and iii) provided a new image of better quality.

  1. Line 292 Figure 5, line 314 Figure 6: Please involve at least a brief description of the statistical method in the figure legend.

As kindly requested by the Reviewer, we added a brief description of the statistical method in the figure legend.

  1. Lines 306-309: It’s OK to use the previous study to support current research. But the reviewer doesn’t think it’s a good idea that the authors want to conclude the current TNBC study using previous myeloid cells results.

As kindly suggested by the Reviewer, we have removed our previous data in the last part of the discussion section.

  1. IL-3 is one of the important pro-inflammatory signaling factors and cytokines, there is no surprise that the receptor IL-3Rα is overexpressed in TNBC. The authors provide HIC results that IL-3Rα increased in clinical tissue samples, and TNBC cell lines to suggest IL-3Rα is a therapeutic target. Are there any therapeutic data and deep insight results to show? Please provide.

We thank the reviewer for His/Her comment. We have previously demonstrated that IL-3Rαblockade on tumour endothelial cells (TEC) changes the cargo of the released extracellular vesicles (anti-IL-3R-EV) impacting on tumour progression. A similar antibody-based approach is under investigation. However so far, we have preliminary data, therefore we cannot provide further insight on this topic.

Minor Concerns and Comments:

  1. Line 25: Why is “particularly in TNBC”? The reviewer thinks “For example” might be more suitable.

As kindly requested by the Reviewer, we changed particularly with “for example”.

  1. Line 82: “About 163” is not a scientific description.

As kindly requested by the Reviewer, “about” was removed.

  1. Line 150: Please specify the dose of rhIL-3 for each mouse.

As kindly requested by the Reviewer, we specified the dose of rhIL-3.

  1. Line 221: Is there any reason the cut-off age is 55 years old?

We thank the Reviewer for His/Her comment. We arbitrary set the cut-off at 55 years old since we failed to detect differences in the median of age. This sentence has been included in the present version of the Ms (see line 203).

  1. Line 260 Figure 3: Please increase the resolution of the images.

As kindly requested by the Reviewer, we increased image resolution.

  1. I’d suggest the authors include molecular weight in the original images.

As kindly requested by the Reviewer, we included the MW in the original images.

Reviewer 3 Report

The authors have presented an interesting study on the significance of IL-3Ra as a biomarker for TNBC prognosis. They have shown the presence of IL-3Ra in cells, tissues, as well as in the patients samples. There are some concerns regarding the content-

1. English language needs significant improvement.

2. There are very few manuscripts which show the relevance of IL-3 and the only manuscript to state IL-3Ra correlation in TNBC is with the same group as the authors. The authors only found positivity in 60% of the patients which is  not a high number. Also, in the <55 year old patients, Il-3R was only positive in about 44% patients which does  not indicate it as a strong marker.

3. The relevance of IL-3R needs more description in the introduction section.

4. It is  not clear from the results whether IL-3R can be used to predict the stage of cancer and if the levels change with progression

Author Response

We thank the Reviewer for His/Her appreciation of our experimental approach and positive comments.

Comment 1: English language needs significant improvement.

Response: As indicated by the reviewer English language has been revised.

Comment 2: There are very few manuscripts which show the relevance of IL-3 and the only manuscript to state IL-3Ra correlation in TNBC is with the same group as the authors. The authors only found positivity in 60% of the patients which is not a high number. Also, in the <55 year old patients, Il-3R was only positive in about 44% patients which does  not indicate it as a strong marker.

Response: We agree with the Reviewer that no other papers provide direct evidence on this topic. We thought it would represent a novelty.

In Table 1 we have reported that 69% of patients are positive for the IL-3Rα expression and that 56% of >55y old patients also expressed IL-3Rα. Moreover, different pathologists using a diverse batch of antibody validated its expression (68.5%) in a second centre (Sassari). We have considered the validation as a benefit.

Comment 3: The relevance of IL-3R needs more description in the introduction section.

Response: As kindly suggested by the Reviewer a paragraph reporting the relevance of IL-3Rα has been included in the present version of the Ms.

Comment 4: It is not clear from the results whether IL-3R can be used to predict the stage of cancer and if the levels change with progression.

Response: We thank the Reviewer for this appropriate request. As shown in the Table the majority of TNBC referred to G3 stage, however the percentage of IL-3Rα positive samples at G2 was 15% compared to the 8% in the IL-3Rα negative group. Due to the n of G2 samples, we cannot drawn conclusions on the ability of IL-3Rα expression to predict the tumour stage. We are now performing a prospective study with the aim to include  patients at the early stage of disease. We hope that additional information would be obtained. In addition, if it would be possible, the level of IL-3R expression will be analysed and compared in samples from the same patients at the diagnosis and recurrence.

Reviewer 4 Report

general comment

-The paper is highly interesting and could have important implications for TNBC research. 

- The method is properly explained and conducted properly.

-The data is well-organized and demonstrated.

-the result is clearly displayed

specific comments

- combine line 55 and 58-59: discussing on metastasis

- Better to note the staging of nodal involvement (N2) on line 230. 

-It would be better to revise the caption on Figure 1 because it does not adequately describe the images.

-Figures 2 and 3's captions need to be revised since they don't adequately describe the figures.

- Figure 4 A's administration to mice is every two days, but the caption states that it is twice a week. This is confusing and has to be expressed consistently.

- Figures A–H are not clear, and it would be preferable to substitute them for better-quality ones or enhance their resolution.

- A brief discussion paragraph that demonstrates the heterogeneity in IL-3Rα  responses on distinct triple negative breast cancer subtypes and their implications in terms of clinical significance brings attention to this paper.

Author Response

We thank the Reviewer for His/Her appreciation of our experimental approach and positive comments.

Comment 1: combine line 55 and 58-59: discussing on metastasis

Response: As kindly suggested by the Reviewer, we have combined the indicated lines.

Comment 2: Better to note the staging of nodal involvement (N2) on line 230. 

Response: As kindly suggested by the Reviewer, N2 has been included in the main text.

Comment 3: It would be better to revise the caption on Figure 1 because it does not adequately describe the images.

Response: As kindly suggested by the Reviewer, caption in Fig.1 has been revised

Comment 4: Figures 2 and 3's captions need to be revised since they don't adequately describe the figures.

Response: As kindly suggested by the Reviewer, captions in Fig.2 and 3 have been revised

Comment 5: Figure 4 A's administration to mice is every two days, but the caption states that it is twice a week. This is confusing and has to be expressed consistently.

Comment 6: Figures A–H are not clear, and it would be preferable to substitute them for better quality ones or enhance their resolution.

Response: As kindly suggested by the Reviewer Figure 4 has been divided into two figures thereby ameliorating the quality. Moreover, the protocol workflow has been revised in Figure 6 accordingly with the Reviewer suggestion.

Comment 7: A brief discussion paragraph that demonstrates the heterogeneity in IL-3Rα responses on distinct triple negative breast cancer subtypes and their implications in terms of clinical significance brings attention to this paper.

Response: We really thank the Reviewer for His/Her suggestion. A comment of the heterogeneity of IL-3 response in TNBC has been included in the discussion section.

“In addition consistent with ‘omics technologies’ [55], our results provide evidence on the heterogeneity of IL-3Rα response in different TNBC (the ability to induce the expression of EMT marker was tumour specific) as well as on the ability of the IL-3Rα to activate com-mon pathways (the induction of the epithelial-to-endothelial switch was detected in all the cell lines tested).”

Round 2

Reviewer 1 Report

The authors have addressed my concerns. The revised version reads well. I recommend acceptance.

Author Response

We thank the Reviewer for His/Her helpful advices.

Comment 1: Inappropriate self-citations did not remove.

Response: As suggested by the Reviewer previous Ref. 12 and 35 were removed

Comment 2: Only 3 mice in each group are not enough.

Response: We are confident that an additional group of animals will improve our results, however it would take more than 3 months to perform all the protocol and we have only 10 days to revise the Ms.  

Comment 3: Line 264: The Figure 4 images are very hard to check, especially Figure 4A to 4H.

Response:

As suggested by the Reviewer Figure 4 has been divided into two figures thereby ameliorating the quality.

Comment 4: The original images did not update.

Response: We apologise, in the present version of the Ms the original wb has been correctly uploaded.

Comment 5: There is no deep insights investigation to prove IL-3Rα can be used as a valuable therapeutic target in the manuscript. 

Response: According with the time of revision we decided to renew the title removing the potential application of IL-3Rα blockade as therapeutic. The main text has been revised accordingly.

Reviewer 2 Report

Thank you for updating the manuscript. However, the reviewer does not think the manuscript has been fully improved.

1) Inappropriate self-citations did not remove.

2) Only 3 mice in each group are not enough.   

3) Line 264: The Figure 4 images are very hard to check, especially Figure 4A to 4H.

4) The original images did not update.

5) There is no deep insights investigation to prove IL-3Rα can be used as a valuable therapeutic target in the manuscript. 

Author Response

(The authors gave the same response as above.)
